# Mountain moves: Spatial interaction modelling of Bulgaria's internal migration (1934-1992)

Petrus J. Gerrits[1]*, Guy Solomon[2], M. Erdem Kabadayi[3], Ana Basiri[1]

1 School of Geographical and Earth Sciences, The University of Glasgow, Glasgow, United Kingdom,
2 School of History, Philosophy and Digital Humanities, The University of Sheffield, Sheffield, United Kingdom, 3 Department of History, Koç University, Istanbul, Turkey

* p.gerrits.1@research.gla.ac.uk

## Abstract

Bulgaria experienced dramatic rural decline alongside rapid urban growth during the 20th century, shaped by both demographic pressures and socioeconomic change. Today, it remains one of Europe's fastest-declining populations, underlining the importance of understanding long-term migration dynamics. Understanding these migration dynamics is essential for interpreting the country's broader population shifts. This study provides a spatial analysis of internal migration in Bulgaria from 1934 to 1992. We construct a harmonised geocoded census settlement dataset, combining historical population records with geospatial settlement boundaries, road network data, and terrain ruggedness measures. Distances between settlements are calculated using both Euclidean and road-network measures, and terrain effects are quantified through terrain ruggedness indices. Migration flows are estimated using spatial interaction models (SIMs), parameterised by population scaling and distance decay functions. Model outputs are validated against historical benchmarks and aggregated regional flows, as well as on the settlement level, by intercensal period variability, ensuring robustness between the intercensal periods. Our analysis investigates the role of challenging topography in shaping migration flows, showing how mountainous landscapes constrained movement while facilitating concentrated urban growth. By integrating historical census records with spatial modelling and geospatial analysis, we uncover local migration dynamics that remain invisible at larger scales. Although our study does not offer direct policy advice, it provides a quantified geospatial perspective on historical context for contemporary policy debates and urban planning initiatives in a country that has experienced both significant rural decline and rapid urbanisation. The findings shed new light on Bulgaria's population history and provide a framework for understanding the interplay between landscape features and migration dynamics.

**Data availability statement:** The complete scripts, including data processing and SIM calibration, and maps, are available in a public GitHub repository (https://github.com/pjgerrits/bg_historical_sim) and Zotero (DOI: 10.5281/zenodo.15309557).

**Funding:** Funding support was provided by the UrbanOccupationsOETR European Research Council (UrbanOccupationsOETR, grant agreement ID: 679097 and GeoAI LULC Seg, grant agreement: 101100837, PI: M. Erdem Kabadayi), and by the UK Research and Innovation Future Leaders Fellowships (grant reference: MR/Y011856/1 and MR/S01795X/2, PI: Ana Basiri). The funders had no role in study design, data collection and analysis, decision to publish, or preparation of the manuscript.

**Competing interests:** The authors have declared that no competing interests exist.

## Introduction

Over the past century, Bulgaria has undergone far-reaching demographic transformation, characterised by sustained rural depopulation and rapid urban expansion [1]. During the second half of the twentieth century, rural populations declined sharply while the proportion of residents living in towns and cities increased markedly [2–5]. These shifts, driven by industrialisation, socioeconomic restructuring and geopolitical change, have left lasting imprints on Bulgaria's social and environmental landscape [6]. Yet despite extensive research on the demographic outcomes of this transformation, much less is known about its spatial mechanisms and quantitative dynamics: how internal migration evolved across settlements and regions and how geographic factors shaped these movements over time. By adopting a quantitative spatial perspective, this study reconstructs and analyses long-term internal migration patterns to reveal the processes that underpinned Bulgaria's twentieth-century population change. Today, as the country remains one of Europe's fastest-declining populations [7], such historical understanding provides essential context for interpreting its contemporary demographic trajectory and offers insights relevant to other regions facing similar spatial demographic and challenges worldwide.

Internal migration reshapes communities by altering settlement patterns, driving demographic change, and stimulating regional development [8]. Such population redistribution not only reflects broader socio-economic transformations but also responds to infrastructural investments and environmental constraints. However, studying internal migration is challenging because data are often scarce or of uneven quality [9]. Previous studies on Bulgarian internal migration have primarily relied on aggregated data at higher administrative levels—namely, the 28 districts (*okrags* before 1987, *oblasts* after) or the 265 municipalities (*obshtini* today) [2,3,6,10,11]. By integrating geocoded historical gazetteer records with national census data in a multi-disciplinary geospatial framework, this study uncovers novel insights into Bulgaria's internal migration dynamics. Our research goes beyond these administrative levels and investigates the population dynamics on the settlement level, analysing 5302 settlements (i.e., EKATTE, Unified Classification of Administrative-Territorial and Territorial Units) with population values over time (see Fig 1). According to the Bulgarian National Statistical Institute (NSI), as of the start of 2024, Bulgaria counted 5256 settlements, of which 257 were towns and 4999 were villages [12]. To our knowledge, there have been no studies examining population dynamics at this administrative level for the analysed time period.

In this study, we analyse inter-settlement migration in Bulgaria using geocoded settlement-level census data from the NSI combined with a spatial interaction modelling (SIM) approach. This granular perspective not only investigates the local drivers of migration but also sheds light on the relationships between settlement structures and hierarchies, providing new insights into the country's demographic dynamics and environmental constraints. Our analysis builds on recent advances in spatial interaction modelling applied to internal migration and human mobility, as demonstrated in prior studies [13–19], and extends these methods to a historically nuanced setting in Bulgaria. However, prior studies on historical

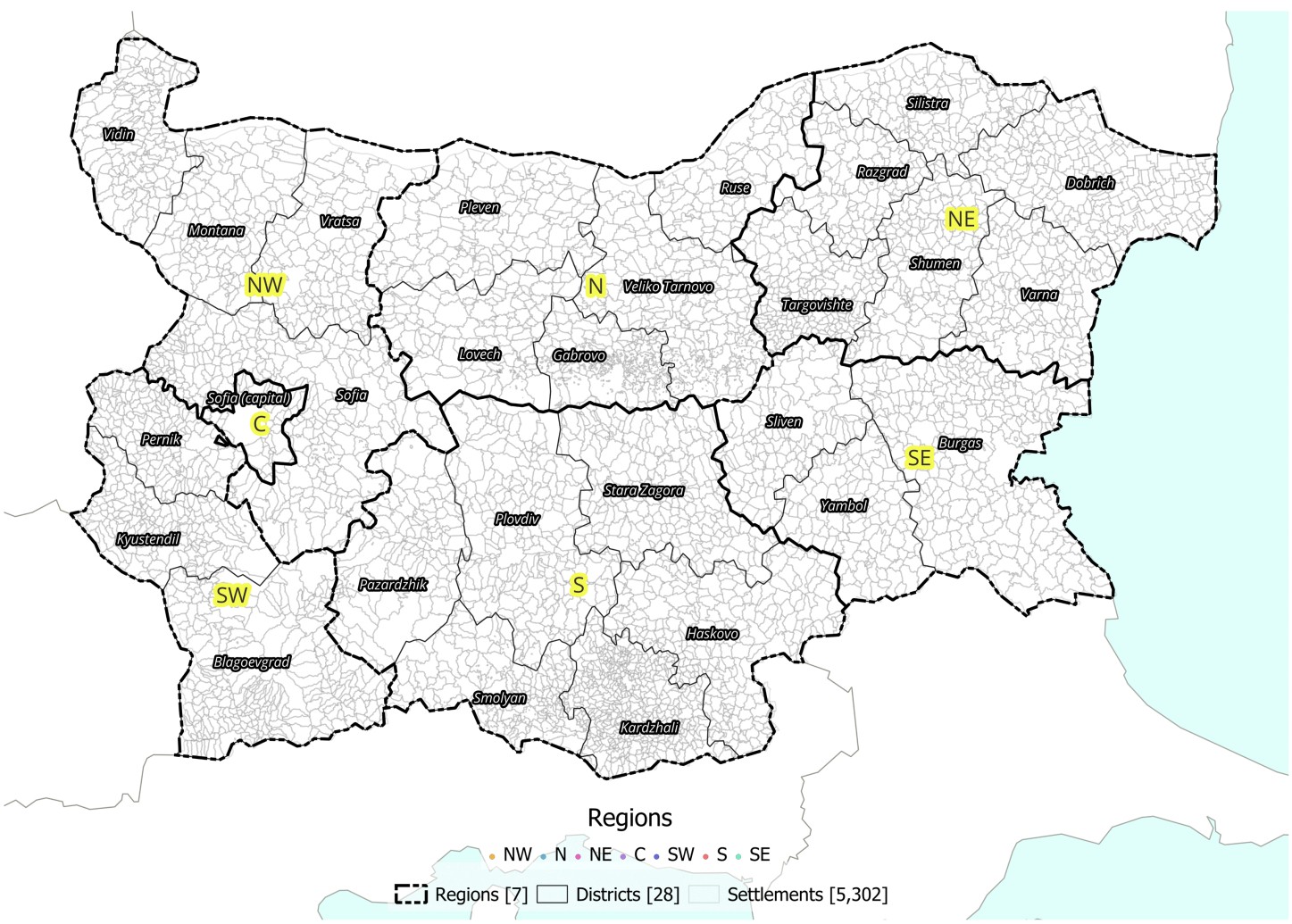

**Fig 1**. **Map overview of the study area, showing the 5302 settlements areas, 7 aggregated regions and 28 districts used in this research.**

demography of Bulgaria mostly lack the spatial context, and focus on depopulation after the 1990s, few studies on historical population geography of the country either do not have a national coverage or high spatial resolution compared to our study [20,21]. One of the key advantages of our higher spatial resolution approach is its ability to integrate environmental factors into the analysis of migration flows.

The period of our examination spans 1934 to 1992, and it is based on all seven population censuses conducted within these years. Beginning with the 1934 census conducted after the 19 May coup d'état, which ended the rule of the Popular Block, consisting of the Democratic Party and the Bulgarian Agrarian National Union, and resulted in the consolidation of the monarchic rule of Boris III between 1935 and 1943. The 1934 national census conducted at the end of the year marks the last data point before World War II, with its substantial impacts on the politics and demography of the country, including the establishment of communist rule in 1946. The 1992 census, on the other hand, is the first one conducted after the end of the People's Republic of Bulgaria as a socialist republic in 1990, capturing the demographic effects of the 1989 "Big Excursion" forced migration of Bulgarian Muslims to Turkey [22]. This period therefore encompasses Bulgaria's major political transformations, from the pre-war monarchy through the socialist regime to the onset of democratic transition. The

decision to restrict our analysis to the period ending in 1992 is based on several important considerations. Most importantly, the opening of Bulgaria's borders in 1989–1990, following the fall of the Berlin Wall, led to a marked shift in migration behaviour and data collection practices. Additionally, the post-1992 era is characterised by rapid political and economic transformations that introduce inconsistencies, challenging methodological uniformity [6,23]. Our analysis therefore concentrates on 1934–1992, a period characterised by administrative consistency in census taking and relative political stability, including the immediate periods before and after the Communist rule in Bulgaria.

The extended timeframe of our analysis provides a unique longitudinal perspective on Bulgaria's urban development. Before 1944, the country's urbanisation rate remained relatively low, reflecting the slow pace of industrialisation and the predominance of rural livelihoods. This pattern changed dramatically during the socialist period (1944–1989), when accelerated industrialisation and state-led urbanisation policies transformed the national settlement system. As Koulov observes, the number of officially designated towns increased by 243 percent between 1946 and 1985, with 134 new towns declared. These changes not only reshaped the settlement hierarchy but also redefined the role of small towns, many of which served as labour reservoirs for large industrial projects [1]. According to Philipov, Bulgaria's urban population increased from 19.9% of the total in 1900 to 21.4% in 1934 and reached 58.7% by 1975. Prior to 1944, Bulgaria's urbanisation rate was relatively low, partly because of a slow industrialisation process [5]. This demographic shift was driven not only by internal migration but is also by declining reproductive rates and ageing rural populations, reflecting broader demographic transitions. The rapid urbanisation that followed has had enormous effects on the spatial and social structure of the country, leaving enduring imprints on its demographic geography.

Building on this historical context, our analysis places particular methodological emphasis on topographic factors, examining how the rugged terrain of mountainous regions affects migration dynamics relative to other parts of Bulgaria. Bulgaria's mountainous and border regions are frequently highlighted as being among the most heavily affected by depopulation [2]. Nearly 50% of the country is classified as mountainous, and since 1975 these areas have experienced a more rapid population decline compared to other regions [24]. The Bulgarian state has been aware of this trend and has implemented targeted measures—such as the "Strandja-Sakar" program in the 1980s—aimed at reversing depopulation in these areas. However, geographic problems of policy formation have hindered such initiatives and proved challenging [1]. By focusing our quantitative analysis on Bulgaria's mountainous regions, we assess their relative attractiveness compared to larger urban centres and other non-mountainous zones. Furthermore, the integration of historical census data with modern geospatial techniques—as demonstrated in studies of agricultural land abandonment [25–27]—shows that fine-resolution analyses can capture long-term trends often overlooked at broader scales. While there are studies focused specifically on mountainous regions in Bulgaria, none have been conducted at a national scale using settlement-level data [28].

In light of these historical and methodological considerations, this study addresses a critical gap in Bulgarian spatial demographic research by providing a spatially explicit account of internal migration at the settlement level. Its main objective is to capture both the temporal evolution of internal migration intensity and the influence of topographic and regional constraints on population movement. To achieve this, the study makes three key contributions. Firstly, it applies spatial interaction models to historical census data, moving beyond traditional district-level analyses to offer a detailed representation of urban–rural migration dynamics across six decades. Secondly, it provides new insights into Bulgaria's long-term demographic and socio-economic transformations, highlighting how physical geography and infrastructural development have jointly shaped migration flows. The specific focus on mountainous regions underscores the importance of fine-grained spatial demographic analysis in environmentally constrained contexts. Lastly, it demonstrates the value of integrating modern geospatial techniques—including OpenStreetMap-based road network data—with harmonised historical census records, establishing a spatially explicit framework for analysing historical migration processes within both temporal and environmental dimensions.

## Materials and methods

This study applies a spatial interaction modelling (SIM) framework to settlement-level census data to examine internal migration flows in Bulgaria between 1934 and 1992. It is ordered in the following way: (i) data acquisition, (ii) data processing, (iii) analysis (spatial interaction modelling), and (iv) validation followed by the results. A schematic workflow is provided in Fig 2 to summarise the data inputs, processing steps, spatial interaction modelling, and validation steps in more detail.

### Data acquisition

To capture the many aspects of internal migration, we rely on several complementary datasets—settlement-level population records, coordinates of settlement points, a detailed OSM-based road network, and environmental data capturing terrain variability. This section gathers the sources exactly as used in the study, before any processing or analysis is conducted.

**Population data.** Population data were obtained from the Bulgarian National Statistical Institute website, which provides settlement-level counts along with basic socio-demographic characteristics and administrative classifications (e.g., villages, municipalities and districts). This dataset, which covers the consistent period from 1934 to 1992, has been widely used to study Bulgarian urbanisation and rural depopulation [2,10]. Population figures for the years 1934, 1946, 1956, 1965, 1975, 1985, and 1992 were considered in the analysis, as displayed in Table 1.

**Geospatial data.** To ensure spatial accuracy, settlement locations and areas were georeferenced using latitude and longitude coordinates provided by the Bulgarian Agency for Geodesy, Cartography and Cadastre (https://kais.cadastre.bg). Official EKATTE codes, i.e. Bulgarian place identifiers, were used to match these geospatial records against the population data, providing the basis for the construction of reliable distance matrices between settlement points. Using EKATTE as the common identifier enables consistent joins on settlements across years and sources. The final list of settlements, coordinates and EKATTE identifiers was shared and publicly available in the data availability statement.

**Road network data.** Recognising that real-world travel distance is rarely captured by straight-line distances, we computed a road distance matrix based on OpenStreetMap (OSM) data [29], downloaded from Geofabrik (https://download.geofabrik.de/europe/bulgaria.html). These distances were calculated for each origin–destination pair using the Open Source Routing Machine (OSRM) Python package [30]. Given the lack of digitised, national-level data for travel networks during the period of study, OSM road network data is selected as a proxy. This method accounts for actual travel routes, infrastructure constraints, and terrain elevation and slope variability, offering a more realistic measure of the "friction of distance" between settlements. Fig 3 illustrates the comparison between OSRM-based pairwise distances and Euclidean distances. Naturally, in all cases, the OSM-based distances are longer than the straight distance between a point of origin and destination.

**Topographic ruggedness data.** In addition to these demographic and infrastructural datasets, we also incorporated environmental information to assess the impact of terrain on migration. For this purpose, we derived the Topographic Ruggedness Index (TRI) from 30m-resolution SRTM (Shuttle Radar Topography Mission) data. TRI has been widely used to analyse regional differences in rugged terrain (e.g. the work by Nunn and Puga to analyse the long-term impact of terrain in Africa [31]). To calculate the TRI value, we followed the methodology first implemented by Riley et al. [32]. There are many alternative approaches and adaptations to investigate the heterogeneity of terrain [33,34], but these are beyond the scope of this paper. The SRTM data [35] were clipped to the Bulgarian boundary using Google Earth Engine [36] (Clipping script is available via Github and Zenodo.). The TRI was then calculated using GDAL's `gdaldem` function [37], which provides a quantifiable measure of ruggedness that may have had a long-term influence on migration dynamics. Fig 4 provides an overview of average settlement-level TRI values in quantile classes.

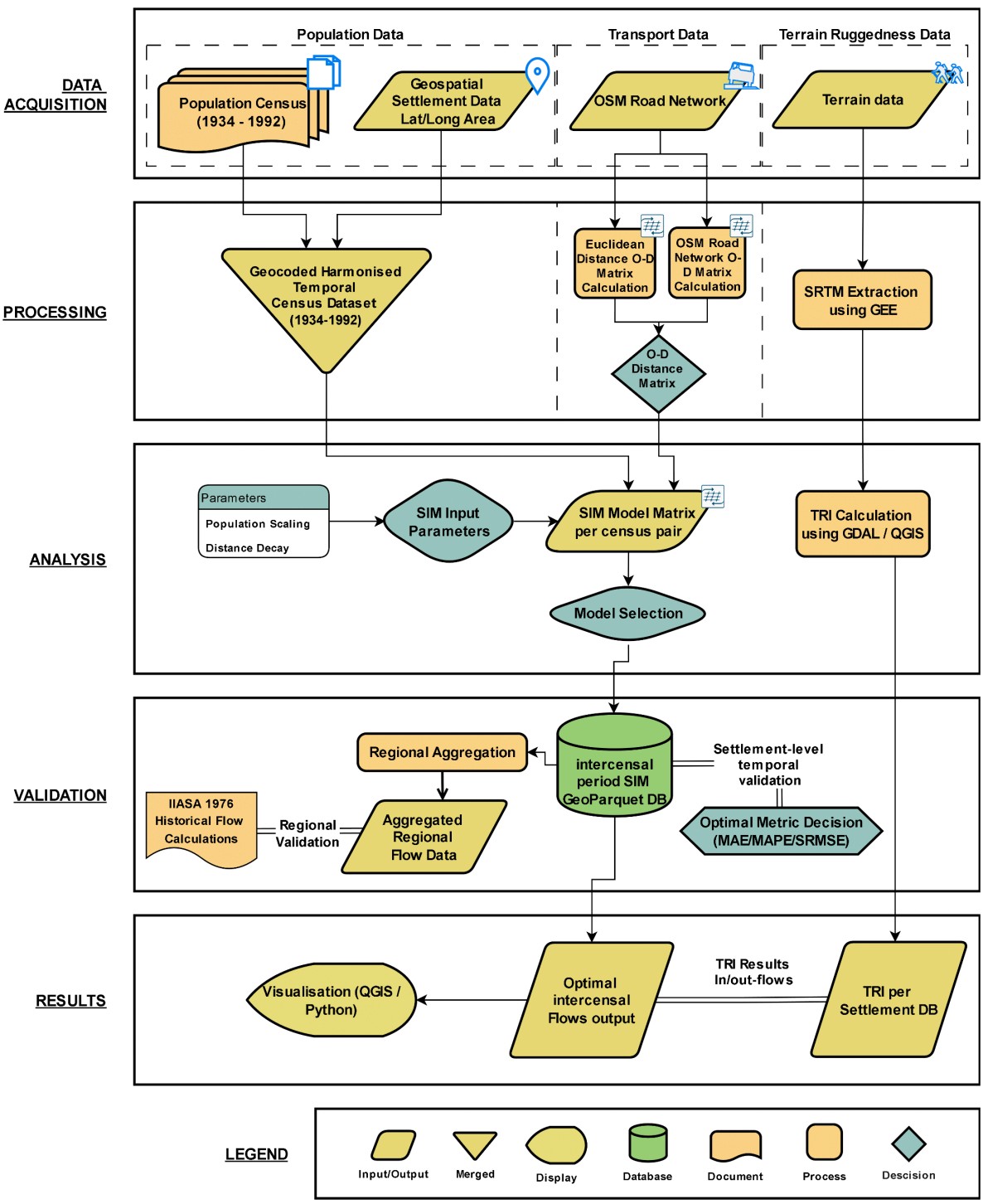

**Fig 2**. Workflow for Bulgarian settlement flow and terrain ruggedness analysis, showing data acquisition, processing, spatial interaction modeling analysis, and validation and the results steps.

**Table 1**. Census years included in this analysis, the count of geocoded settlements per year and population totals.

| Year | Settlements | Population |
|------|-------------|------------|
| 1934 | 5144 | 6,287,453 |
| 1946 | 5157 | 6,953,950 |
| 1956 | 5188 | 7,579,577 |
| 1965 | 5212 | 8,198,334 |
| 1975 | 5217 | 8,710,259 |
| 1985 | 5258 | 8,944,950 |
| 1992 | 5316 | 8,487,317 |

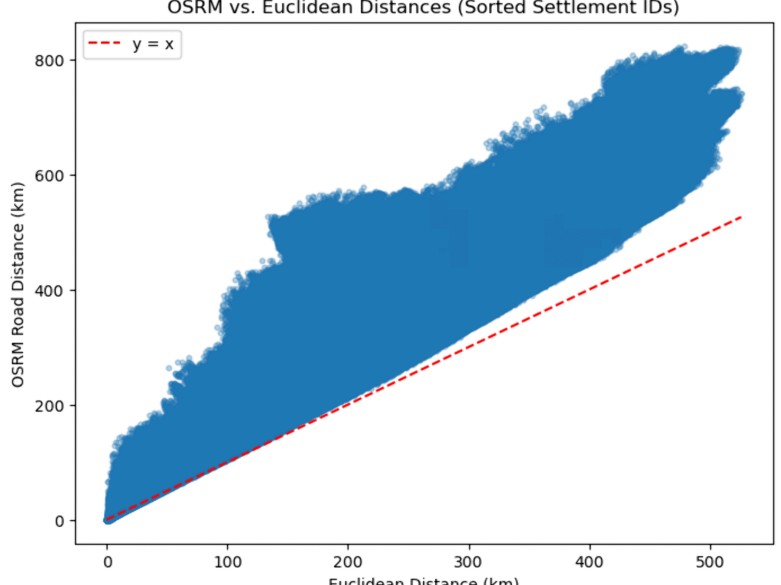

**Fig 3**. Showing the Open Source Routing Machine (OSRM)-based distances on the y-axis, against the Euclidean distances on the x-axis in kilometres for every origin–destination pair in our model.

## Data processing

The data processing stage involved several key steps to integrate, refine, and validate the diverse datasets for analysis. First, population records, geospatial coordinates, and the road network distance matrix were harmonised using settlement names and official EKATTE codes, ensuring alignment between demographic census and spatial data. Any discrepancies were resolved by cross-referencing with official NSI publications and historical cartographic records. This harmonisation was undertaken concurrently with the Ottoman Gazetteer project, within the framework of the *UrbanOccupationsOETR* ERC project [38–40]. In addition to record linkage, population tables were screened for inconsistencies, with missing or anomalous values verified against NSI sources. Settlement names were standardised to match geospatial records, addressing issues such as alternate spellings and boundary changes based on authoritative references.

The pairwise road distance matrix between each settlement pair was computed from OpenStreetMap (OSM) data using the OSRM Python package [30]. After calculating travel distances between each origin–destination (OD) pair, the distances were compared with the Euclidean distance matrix to validate plausibility and to highlight the systematic elongation of network travel relative to straight-line distance. For settlement pairs which were not directly connected through a road

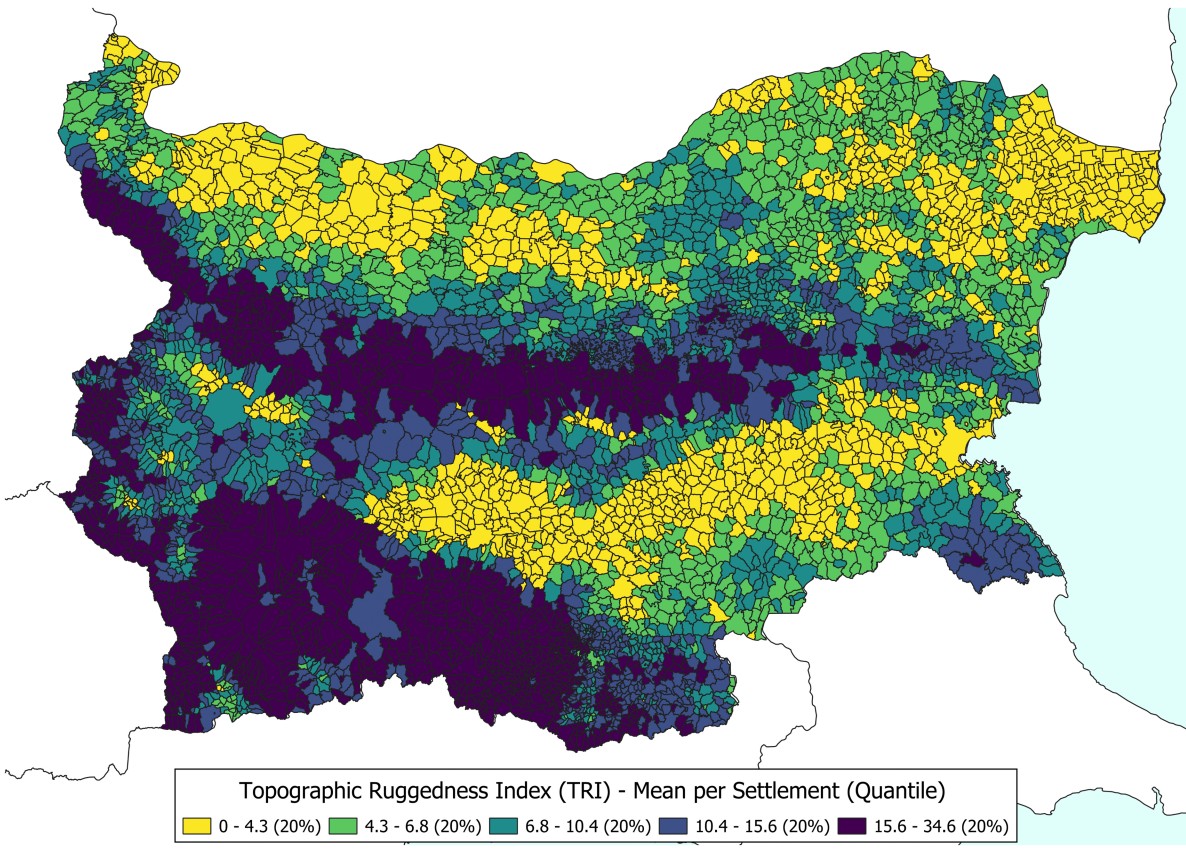

**Fig 4**. **Overview of Bulgaria's mountainous areas by average TRI value per settlement.** Higher TRI values are considered more rugged areas in the country. Settlements are classified in 5 categories in a quantile distribution.

connection, the closest main road was used to calculate the route, thereby ensuring realistic and topologically valid paths rather than imposing an arbitrary large distance.

In addition, environmental data were processed to derive the TRI measurements at the settlement level. We obtained 30m-resolution SRTM data and clipped it to the Bulgarian boundary using Google Earth Engine. TRI values were then calculated using GDAL's `gdaldem` function, providing a quantitative measure of terrain variability that may influence migration patterns. Finally, the cleaned and validated datasets—comprising population figures, geospatial coordinates, road distances, and environmental indicators—were merged into a Parquet database that could be more easily queried. This preprocessing pipeline establishes a consistent foundation for our subsequent spatial analysis.

## Analysis

Spatial interaction models (SIMs) have long served as a cornerstone for analysing migration flows by decomposing movement patterns into components such as origin population, destination attractiveness, and distance decay. Modern spatial interaction model approaches were founded on the early work of Wilson [41] and Roy [42]. Subsequent research, such as that conducted by Bijak [16] and Ramos [15], showed that SIMs were effective in predicting migration trends. More recent applications have refined these methodologies for internal migration analysis in various contexts—from European interregional migration [17,18] to studies in Sub-Saharan Africa [43] and beyond [44]. Advanced formulations such

as kernel-based approaches [45] and non-Gaussian analyses [46], alongside deep learning frameworks [47,48], further highlight the evolving capacity of SIMs to capture the complexity and directionality of migration flows [49].

In the context of modelling long-term internal migration, Wilson's entropy-maximising framework [50] yields four principal spatial interaction models. The *unconstrained model* predicts migration flows solely as a function of origin mass, destination attractiveness, and a distance-decay term, without enforcing any observed total flows on a particular zone. The *production-constrained* model fixes each origin's total out-migration to match empirical counts, permitting destination inflows to adjust freely. Conversely, the *attraction-constrained* model holds each destination's total in-migration constant while allowing origin outflows to vary. Finally, the *doubly-constrained* model simultaneously enforces both origin and destination marginals, ensuring that predicted flows exactly satisfy known origin totals and destination totals. Collectively, these models provide a versatile toolkit for analysing the dynamics of long-term internal migration, allowing researchers to select the appropriate level of constraint in line with their empirical data and theoretical objectives.

In this study, we employ an *unconstrained* (or total constrained) spatial interaction model that draws on the theoretical basis of gravity models [18,51,52]. The choice of an unconstrained model is motivated by our objective to capture the magnitude of migration flows without imposing fixed totals on either the origin or destination side, facilitating a more direct interpretation of the underlying spatial dynamics based directly on the census source material. Following the formulation introduced by Wilson [50], migration flows, $F_{ij}$, between an origin $i$ and a destination $j$ are modelled as follows:

$$F_{ij} = k \times P_i \times A_j \times f(d_{ij}), \tag{1}$$

Where $P_i$ represents the population at origin $i$, $A_j$ denotes the attractiveness of destination $j$, $f(d_{ij})$ is a distance decay function capturing the impedance effect of separation between settlements, and $k$ is a scaling constant. This formulation enables the disentanglement of the effects of population size, distance, and other socio-economic variables on migration flows. Two distance-decay formulations were considered in this analysis: an inverse power model and an exponential model, reflecting sensitivity to long- versus short-distance movements.

For each year pair in our analysis, we systematically calibrated our spatial interaction model through a grid search over key input parameters. In particular, we varied the population scaling factor ($\gamma$) from 1.2 to 1.9 in increments of 0.1, which modulates the influence of the destination population on the attractiveness of settlements. Concurrently, we iterated the distance decay parameter ($\beta$) from 0.5 to 3.0 for both distance-decay formulations.

In the inverse power formulation, migration flows $F_{ij}$ are expressed as

$$F_{ij} = k \times P_i \times (P_j)^{\gamma} \times d_{ij}^{-\beta}, \tag{2}$$

where $k$ is a scaling constant, $P_i$ and $P_j$ are the origin and destination populations, and $\beta$ controls the strength of the distance-decay effect.

In the exponential formulation, the relationship becomes

$$F_{ij} = k \times P_i \times (P_j)^{\gamma} \times \exp(-\beta d_{ij}), \tag{3}$$

where the exponential term reflects a more rapid decline of interaction with distance.

These parameter ranges and model formulations follow the guidelines outlined in the Routledge Geospatial Analysis Handbook [53], ensuring that our choices are well grounded in the existing literature. For each census interval, the combination of parameters that best reproduced the observed migration flows was selected for subsequent analyses (see the Validation section for performance assessment).

## Validation

Validating model performance against historical data presents distinct challenges. To ensure robustness, we evaluated model performance using a multi-metric framework that captures both absolute and relative deviations between observed and predicted settlement populations. Specifically, we employed three complementary measures: the mean absolute error (MAE), the mean absolute percentage error (MAPE), and the standardised root mean squared error (SRMSE). Together, these indices provide a nuanced understanding of model accuracy across settlements of different sizes and population magnitudes.

**Settlement-level validation.** To assess the performance of our SIM at the settlement level, we compare predicted population estimates to observed census data. This fine-grained evaluation identifies localised differences and also helps to detect possible biases caused by external factors such as unforeseen economic dynamics or demographic shocks. To ensure a comprehensive assessment, we employ three complementary measures that account for both absolute and relative errors: the Mean Absolute Error (MAE), the Mean Absolute Percentage Error (MAPE), and the Standardised Root Mean Squared Error (SRMSE).

We emphasise the use of these three measures because each captures a distinct dimension of model performance. The SRMSE provides a dimensionless indicator of model fit, allowing comparison across census intervals with markedly different total populations and settlement sizes. This standardisation is particularly valuable in spatial modelling, where absolute errors can be dominated by larger urban centres. Meanwhile, MAE captures the average magnitude of deviations in population counts, while MAPE highlights proportional inaccuracies. Taken together, this multi-metric framework ensures that both large-scale and localised variations in model performance are systematically assessed.

**Mean Absolute Error (MAE).** The MAE quantifies the average of the absolute differences between observed and predicted populations, providing an intuitive measure of model accuracy:

$$\text{MAE} = \frac{1}{n} \sum_{i=1}^{n} \left| P_{i,\text{observed}} - P_{i,\text{predicted}} \right|. \tag{4}$$

Unlike metrics that square errors, MAE treats all discrepancies uniformly, making it robust to outliers and easily interpretable as an average deviation in population counts.

**Mean Absolute Percentage Error (MAPE).** MAPE expresses the average absolute error as a percentage of the observed values, offering a proportional view of prediction accuracy:

$$\text{MAPE} = \frac{100}{n} \sum_{i=1}^{n} \left| \frac{P_{i,\text{observed}} - P_{i,\text{predicted}}}{P_{i,\text{observed}}} \right|. \tag{5}$$

This measure is particularly useful for highlighting discrepancies in relative terms, though it can become unstable for very small observed populations—necessitating careful interpretation in sparsely populated or depopulating settlements.

**Standardised Root Mean Squared Error (SRMSE).** The SRMSE, first discussed by Knudsen and Fotheringham in relation to SIM and model fit statistics as well as included in the popular SpInt Python library for spatial analysis [51,54], provides a dimensionless measure of model fit, enabling direct comparison across census intervals with differing total populations and settlement sizes:

$$\text{SRMSE} = \frac{\sqrt{\frac{1}{nm} \sum_{i=1}^{n} \sum_{j=1}^{m} (F_{ij} - \widehat{F}_{ij})^2}}{\frac{1}{nm} \sum_{i=1}^{n} \sum_{j=1}^{m} F_{ij}}. \tag{6}$$

Here, $F_{ij}$ and $\widehat{F}_{ij}$ denote the observed and predicted migration flows between origin $i$ and destination $j$, respectively, and $nm$ represents the total number of origin–destination pairs. By normalising the root mean squared error by the mean observed flow, SRMSE provides a scale-independent indicator of predictive performance—particularly important in contexts where large urban centres might otherwise dominate aggregate error measures.

Taken together, these three measures provide a balanced framework for evaluating both large-scale and localised variations in model performance. SRMSE ensures comparability across time periods, MAE reflects typical absolute deviations, and MAPE captures relative proportional discrepancies. This multi-metric approach enables a comprehensive assessment of model reliability across Bulgaria's highly heterogeneous settlement system.

Beyond their statistical interpretation, these performance indicators also have important spatial implications. SRMSE highlights the variability of residuals relative to observed values, MAE captures the average magnitude of errors in absolute terms, and MAPE expresses these deviations proportionally—making it possible to evaluate how predictive accuracy varies across spatial and demographic contexts. Employing all three measures together helps to identify potential spatial biases that may arise from aggregation effects or uneven population distributions. For instance, large urban centres can disproportionately influence global error measures (like the RMSE and by extension SRMSE), potentially obscuring systematic outliers in smaller or peripheral settlements. Because migration processes are inherently spatial, shaped by distance, accessibility, and settlement connectivity, the errors themselves are also spatially structured. This connection between spatial configuration and model sensitivity aligns with the insights of Knudsen and Fotheringham [51,54] and Jelinski [55], who emphasise that reliance on a single error metric can exacerbate the modifiable areal unit problem (MAUP), where analytical outcomes vary with the scale or zoning of spatial units [56]. By triangulating SRMSE, MAE, and MAPE, our evaluation explicitly accounts for this spatial context, ensuring that the SIM's predictive performance remains robust across both densely and sparsely populated areas and across multiple spatial resolutions.

The best-performing parameter configurations identified through these metrics form the basis for the analyses presented in the Results section, where Fig 5 visualises the performance outcomes across census intervals.

**Regional validation.** While settlement-level metrics assess local precision, model validation at the regional scale ensures consistency with independently documented migration structures. For this purpose, we use the 1976 and 1981 interregional migration matrix published by the International Institute for Applied Systems Analysis (IIASA) [5]. These data aggregate migration into seven Bulgarian macro-regions—Northwest (NW), North (N), Northeast (NE), Southwest (SW), South (S), Southeast (SE), and Sofia—shown in Fig 1. While this is at an aggregated level, it nevertheless provides us with an external source of validation for our flows.

**Aggregation procedure.** Predicted settlement-to-settlement flows $F_{ij}$ were aggregated into region-to-region totals to form a comparable $7 \times 7$ flow matrix $\widehat{\mathbf{R}}$, where each cell represents the total migration flow from one macro-region to another. This aggregation allows direct comparison between the modelled and observed regional migration structures.

**Flow model metric.** The level of agreement between predicted ($\widehat{R}_{ab}$) and observed ($R_{ab}$) regional flows was quantified using the relative difference:

$$\text{RD}_{ab} = \frac{\widehat{R}_{ab} - R_{ab}}{R_{ab}}, \tag{7}$$

where positive values indicate overestimation and negative values indicate underestimation by the model. When $R_{ab} = 0$, a small stabiliser $\varepsilon$ is applied to prevent division by zero.

This regional validation serves as an independent benchmark for assessing the structural coherence of the aggregated migration patterns produced by the spatial interaction model.

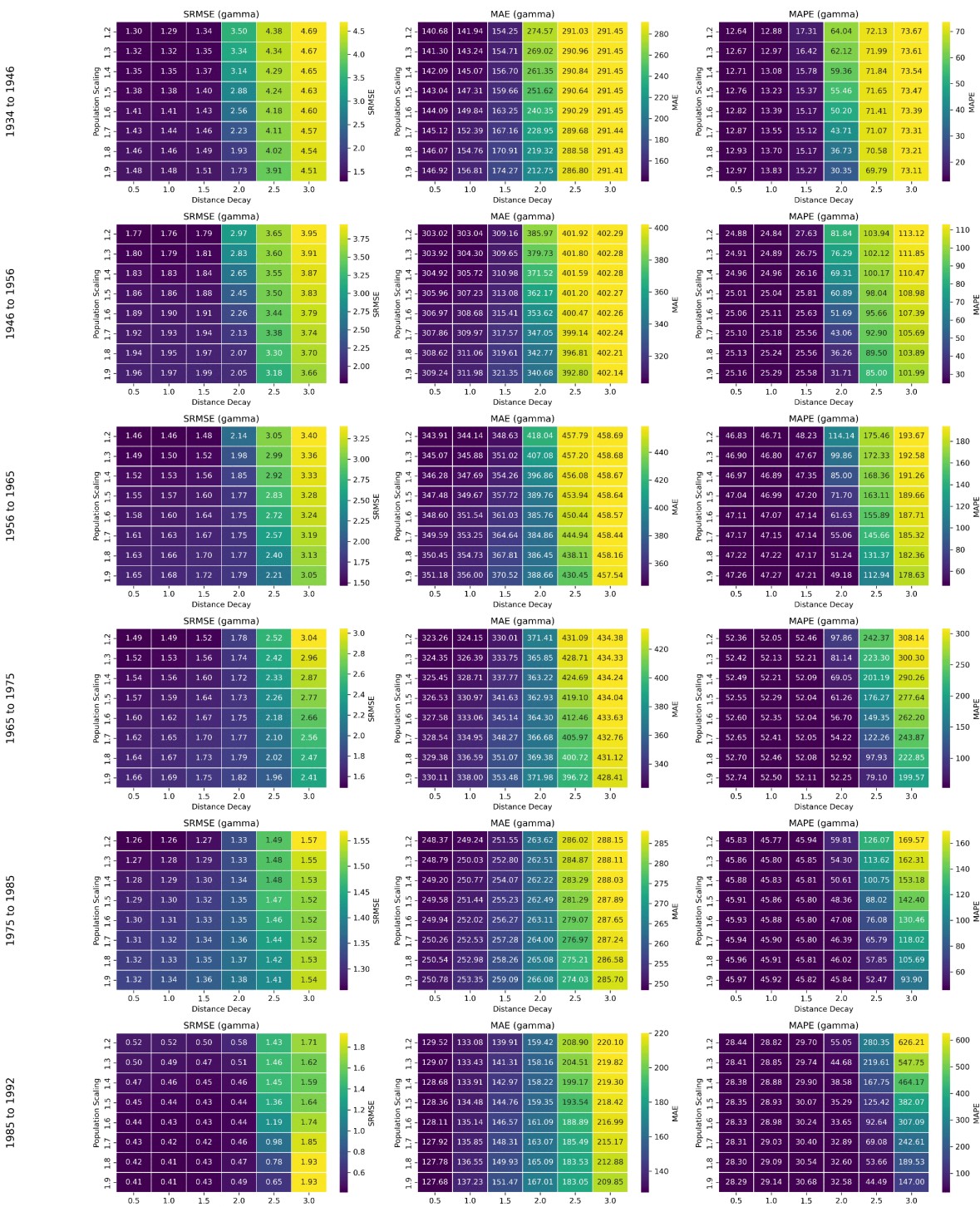

**Fig 5. Settlement-level model performance across intercensal periods (1934–1992) according to SRMSE, MAE, and MAPE values.** Colours range from purple (high accuracy) to yellow (low accuracy).

## Results

### Model validation results

**Settlement-level fit across intercensal intervals (optimal flows output).** An extensive grid search across population-scaling ($\gamma$) and distance-decay ($\beta$) parameters identified the optimal configurations for each intercensal interval. Fig 5 visualises the parameter sensitivity surfaces, and Table 2 summarises the best-performing combinations. Across most periods, moderate distance-decay values ($\beta \leq 1.0$) and lower population-scaling factors ($\gamma \approx 1.2$) yielded the lowest error.

For 1934–1946, the model achieved strong correspondence with observed data (SRMSE = 1.29; MAE = 141.94; MAPE = 12.88). Errors rose during 1946–1956 and 1956–1965, possibly reflecting increased demographic changes in the postwar and early socialist period decades. The 1965–1975 interval shows the highest proportional MAPE error), suggesting that rapid urban-industrial restructuring and policy-driven relocations were not fully captured by the model. Performance improved in 1975–1985 and markedly in 1985–1992 (SRMSE = 0.41; MAE = 137.23; MAPE = 29.14), coinciding with the relaxation of movement restrictions and the shift to a higher optimal scaling factor ($\gamma = 1.9$). This later period is also in accord with possible outcomes of the drastic political change of the fall of communism and the end of movement restrictions.

While SRMSE provides a standardised benchmark across periods, MAE and MAPE offer more interpretable measures of absolute and proportional accuracy. Overall, model performance is consistent and robust across most census intervals, with deviations largely attributable to historical data noise and structural demographic changes rather than model instability.

**External regional flow benchmark (1976).** Using the aggregation and agreement procedures described in the Regional validation section, the model reproduces the broad regional structure of 1976 interregional migration. Totals indicate a modest net underestimation overall (bottom row), with the largest discrepancies concentrated in flows involving the Southwest and Sofia—patterns consistent with their distinctive urban–industrial dynamics in the 1965–1975 window. The undefined relative difference for some Sofia cells reflects zero reported flows in the benchmark matrix rather than model instability. Despite localised deviations, the directional patterning and the ranking of major corridors are captured, providing external support for the settlement-level fit reported above.

### Network visualisation of predicted flows

Our spatial interaction model produced a detailed depiction of internal migration dynamics in Bulgaria between 1934 and 1992. By predicting flows at the settlement level and aggregating these results regionally, we were able to capture both fine-scale migration patterns and broader macro-regional trends.

In our analysis, we retained only flows involving more than one migrant to minimise noise. The resulting network visualisation (Fig 6) clearly identifies major urban attractors, such as Sofia and Plovdiv, which function as hubs in an extensive migration network throughout all decades. These cities not only receive dense inflows from surrounding rural areas

**Table 2**. A combined table showing the best evaluation outputs based on predicted and observed populations based on census records.

| Year Start | Year End | Pop. Scaling | Dist. Decay | Model Type | SRMSE | MAE | MAPE |
|---|---|---|---|---|---|---|---|
| 1934 | 1946 | 1.2000 | 1.0000 | gamma | 1.2916 | 141.9417 | 12.8824 |
| 1946 | 1956 | 1.2000 | 1.0000 | gamma | 1.7612 | 303.0404 | 24.8373 |
| 1956 | 1965 | 1.2000 | 1.0000 | gamma | 1.4628 | 344.1415 | 46.7057 |
| 1965 | 1975 | 1.2000 | 0.5000 | gamma | 1.4880 | 323.2567 | 52.3557 |
| 1975 | 1985 | 1.2000 | 0.5000 | gamma | 1.2586 | 248.3743 | 45.8315 |
| 1985 | 1992 | 1.9000 | 1.0000 | gamma | 0.4105 | 137.2327 | 29.1403 |

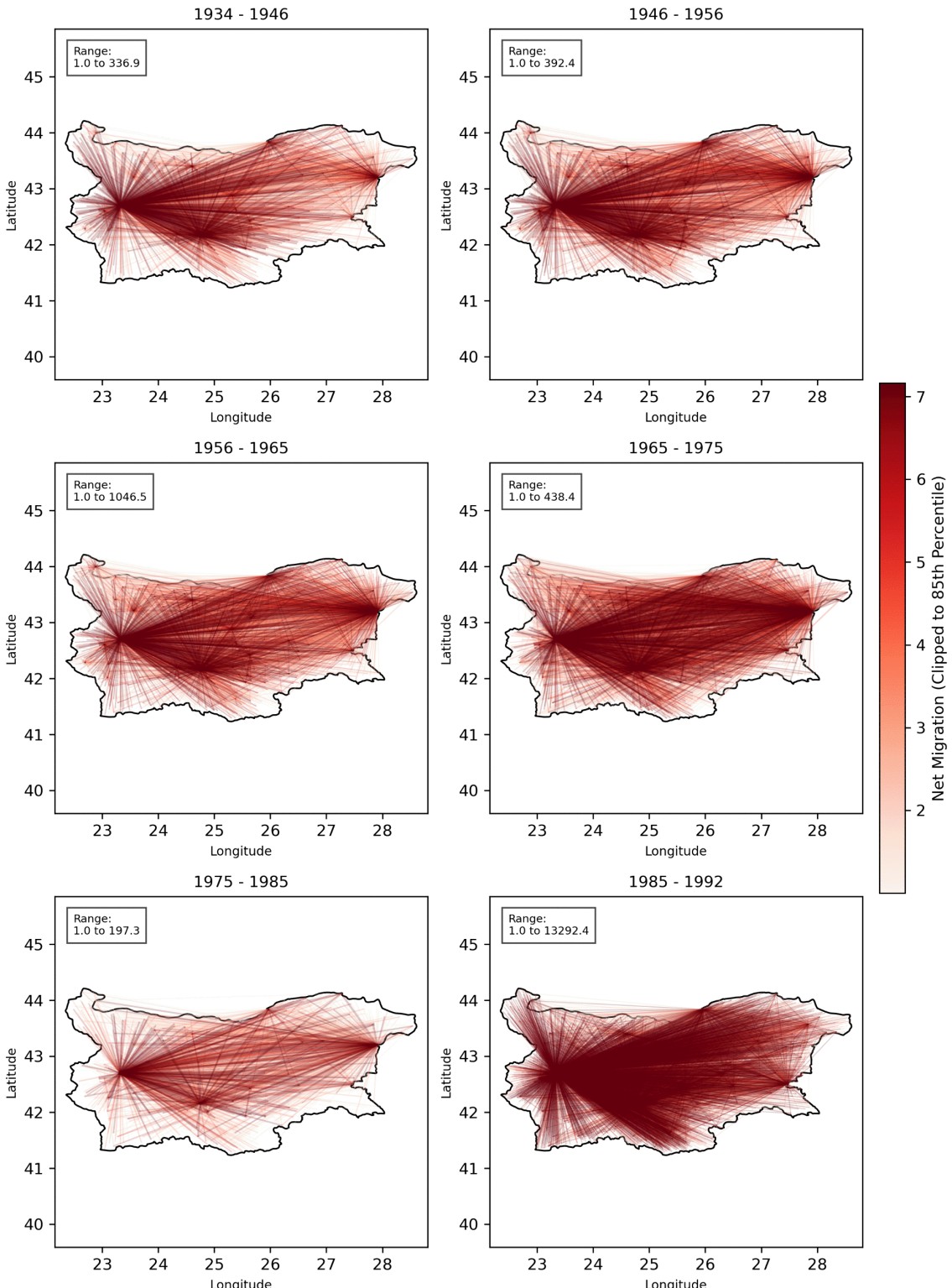

**Fig 6**. **Network visualisation of predicted migration flows between settlements in Bulgaria.** Only flows over 1 are shown, highlighting major urban attractors such as Sofia and Plovdiv.

but also connect multiple regions, reinforcing the long-standing pattern of urban concentration. Secondary centres, including Varna, Burgas, and Stara Zagora, become increasingly prominent after 1956, reflecting the spatial effects of socialist industrialisation and coastal economic expansion.

Across the six intercensal periods, the overall density and spatial reach of migration links increase, suggesting a gradual integration of national migration systems through infrastructure and economic development. By the late socialist period (1965–1985), the visual patterns indicate widespread internal connectivity, with flows extending beyond local catchments to connect multiple regions. The final 1985–1992 interval shows intensified movement toward Sofia and major cities, consistent with the observed rise in total net migration volumes and validation metrics for that period.

Taken together, the larger network patterns on (Fig 6) indicate that the model consistently reproduces the dominant urban attractors observed in the empirical record while remaining responsive to temporal variation in migration intensity. Rather than implying perfect agreement, these results suggest that the model captures the major structural tendencies of Bulgaria's internal migration system, reflecting both the spatial concentration of flows toward large cities and their gradual diffusion through time.

## Intercensal variability of inflows, outflows, and net migration

Beyond the network perspective, settlement-level inflows, outflows, and net migration were computed for each intercensal interval. The statistical distributions of these metrics reveal a consistently right-skewed pattern (Fig 7), with most settlements exhibiting low migration volumes and a smaller subset displaying disproportionately high inflows and outflows. This skewness reflects the uneven spatial distribution of economic opportunities and infrastructure, where urban settlements, particularly regional centres, act as dominant attractors, while rural and mountainous areas experience persistent outflows. The observed intercensal variability in these distributions likely reflects the combined influence of demographic, economic, and infrastructural processes whose individual effects cannot be isolated within the current modelling framework. Further research based on this quantitative spatial movement dataset could therefore provide valuable insights into the specific triggers underlying these fluctuations and potential inconsistencies across intercensal periods. These findings nevertheless reaffirm established long-term demographic trends of rural depopulation in Bulgaria [2].

To further assess the robustness of our model, we aggregated the settlement-level predictions to form regional estimates for Bulgaria's seven macro-regions. A comparison with historical data from the 1976 Bulgarian Statistical Yearbook [57] shows that the overall magnitude and direction of migration flows are well reproduced (Table 3). Slight overestimations of inflows occur in certain regions such as the Southwest, but these discrepancies can be attributed to localised factors that are not fully represented by the baseline model.

A temporal perspective on inflows, outflows, and net migration (Fig 7) further illustrates how distributions shift over time. Despite persistent skewness in each decade, modest changes in the shape and spread of the log-scaled histograms suggest that broader economic and infrastructural developments influenced overall migration levels. The fundamental pattern of a large cluster of lower flows and a thinner tail of higher flows remains evident throughout the intervals, indicating that while the magnitude of migration varies, its skewed nature persists across decades.

For the 1975–1985 period, the histograms (see Fig 7) reveal a notable contraction in the distribution of both inflows and outflows. The density curves indicate that a higher proportion of settlements experienced relatively modest migration flows during this decade, suggesting a period of demographic stabilisation. In contrast to the pronounced surges observed in the 1950s and 1960s, inflow levels in 1975–1985 were more evenly distributed, and outflows remained consistently low. These observations suggest that economic or infrastructural shifts during this period may have moderated migration dynamics, resulting in a more balanced migration pattern compared with earlier decades.

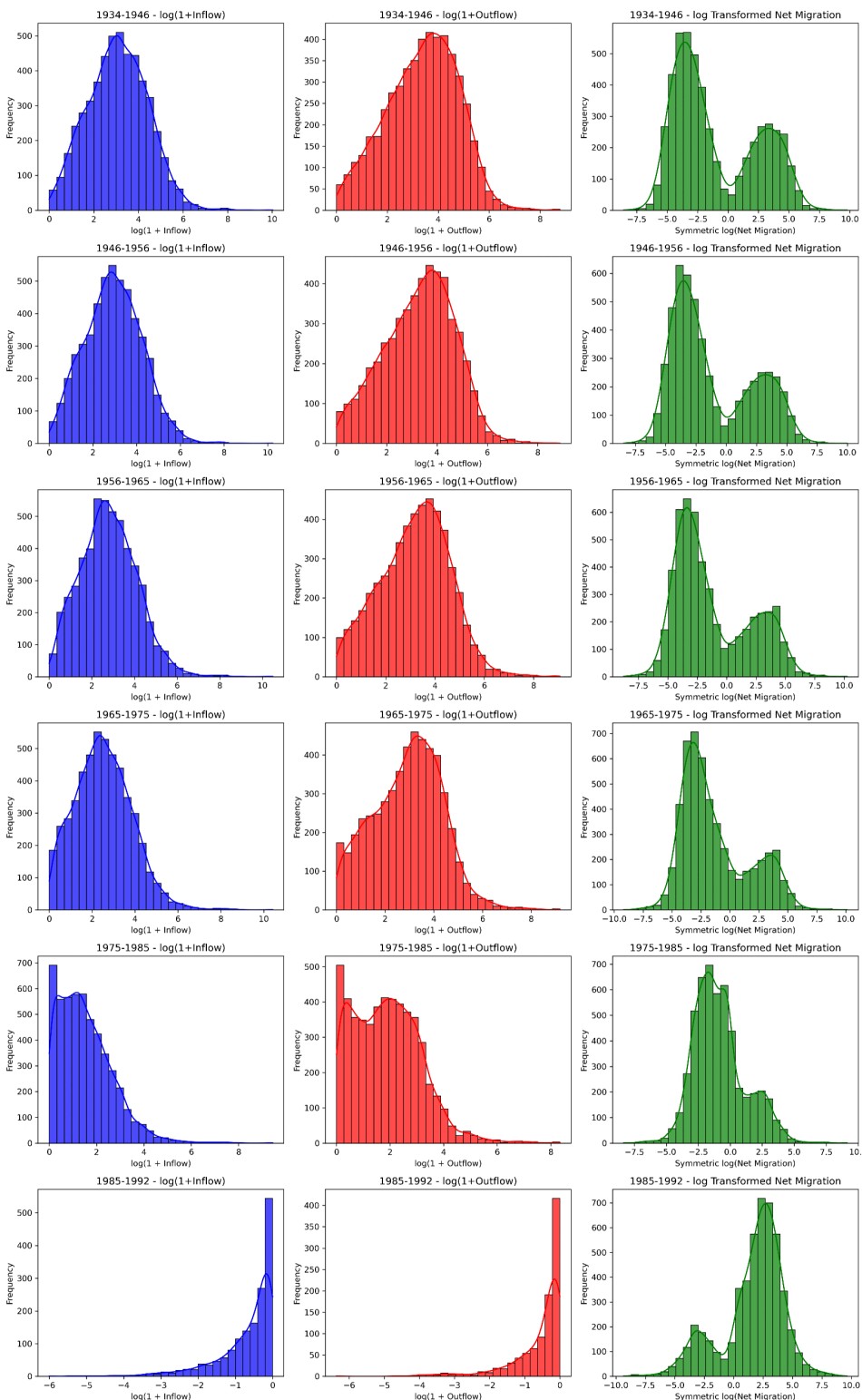

**Fig 7. Histograms and density plots showing settlement-level inflows, outflows, and net migration.**

**Table 3**. **Predicted (model) vs. observed (Philipov, 1976) interregional migration flows aggregated to seven macro-regions; best unconstrained configuration from 1965–1975 applied.** Sofia is listed separately due to its distinct urban dynamics. Bottom panel reports cellwise relative differences (Eq 7); "–" indicates undefined values where the observed flow is zero.

**Predicted flows**

| region_id | region | NW | N | NE | SW | S | SE | Sofia | Total |
|---|---|---|---|---|---|---|---|---|---|
| 1 | **N.West** | 829 | 946 | 693 | 434 | 1282 | 389 | 2233 | 6808 |
| 2 | **North** | 798 | 2028 | 1381 | 479 | 1972 | 747 | 2181 | 9585 |
| 3 | **N.East** | 540 | 1334 | 2037 | 365 | 1571 | 839 | 1490 | 8175 |
| 4 | **S.West** | 382 | 511 | 418 | 475 | 901 | 269 | 1575 | 4531 |
| 5 | **South** | 979 | 1821 | 1559 | 778 | 4609 | 1184 | 3315 | 14245 |
| 6 | **S.East** | 306 | 708 | 848 | 234 | 1226 | 742 | 959 | 5024 |
| 7 | **Sofia** | 701 | 825 | 594 | 585 | 1383 | 389 | 552 | 5029 |
| | **Total** | 4535 | 8172 | 7530 | 3350 | 12945 | 4559 | 12305 | 53397 |

**Flows by Philipov D.**

| | | 1 | 2 | 3 | 4 | 5 | 6 | 7 | Totals |
|---|---|---|---|---|---|---|---|---|---|
| 1 | **N.West** | 1896 | 1042 | 411 | 539 | 1261 | 271 | 1673 | 7093 |
| 2 | **North** | 1175 | 4152 | 2764 | 292 | 1427 | 559 | 747 | 11116 |
| 3 | **N.East** | 471 | 1524 | 4642 | 220 | 983 | 994 | 492 | 9326 |
| 4 | **S.West** | 268 | 146 | 122 | 823 | 298 | 67 | 310 | 2034 |
| 5 | **South** | 854 | 1107 | 759 | 813 | 9766 | 2500 | 1039 | 16838 |
| 6 | **S.East** | 110 | 249 | 502 | 103 | 919 | 1685 | 259 | 3827 |
| 7 | **Sofia** | 3154 | 1446 | 833 | 1987 | 2264 | 864 | 0 | 10548 |
| | **Totals** | 7928 | 9666 | 10033 | 4777 | 16918 | 6940 | 4520 | 60782 |

**Relative Difference**

| | | 1 | 2 | 3 | 4 | 5 | 6 | 7 | Totals |
|---|---|---|---|---|---|---|---|---|---|
| 1 | **N.West** | −0.56 | −0.09 | 0.69 | −0.19 | 0.02 | 0.44 | 0.33 | −0.04 |
| 2 | **North** | −0.32 | −0.51 | −0.50 | 0.64 | 0.38 | 0.34 | 1.92 | −0.14 |
| 3 | **N.East** | 0.15 | −0.12 | −0.56 | 0.66 | 0.60 | −0.16 | 2.03 | −0.12 |
| 4 | **S.West** | 0.43 | 2.50 | 2.43 | −0.42 | 2.02 | 3.01 | 4.08 | 1.23 |
| 5 | **South** | 0.15 | 0.64 | 1.05 | −0.04 | −0.53 | −0.53 | 2.19 | −0.15 |
| 6 | **S.East** | 1.78 | 1.84 | 0.69 | 1.27 | 0.33 | −0.56 | 2.70 | 0.31 |
| 7 | **Sofia** | −0.78 | −0.43 | −0.29 | −0.71 | −0.39 | −0.55 | – | −0.52 |
| | **Totals** | −0.43 | −0.15 | −0.25 | −0.30 | −0.23 | −0.34 | 1.72 | −0.12 |

## Terrain Ruggedness (TRI) effects on in/out-flows and destinations

A novel aspect of this study is the integration of environmental context through the Terrain Ruggedness Index (TRI). The TRI quantifies local topographic heterogeneity by measuring elevation differences between neighbouring cells in a digital elevation model. Unlike elevation or slope alone, which can obscure fine-scale variation, TRI provides a continuous indicator of landscape complexity. Using 30 m SRTM data clipped to Bulgaria's boundary via Google Earth Engine and processed with GDAL's `gdaldem` function, we computed settlement-level TRI values and spatially linked them with migration outcomes across intercensal periods.

The resulting analyses reveal a clear negative association between TRI and absolute net migration (Fig 8). Settlements situated in more rugged terrain tend to exhibit lower net migration values, suggesting that challenging physical geography can act as a constraint to population inflows and may reinforce patterns of outmigration. However, the relationship is not uniform across space or time. The observed variability across intercensal periods reflects the combined influence of demographic, infrastructural, and economic developments, as well as the uneven accessibility of mountainous areas. Although these individual influences cannot be disentangled within the current modelling framework, their combined effect is reflected in the changing spatial distribution and magnitude of migration flows across intercensal periods.

Closer inspection shows that large urban centres, particularly Sofia, exert a strong influence on overall patterns. These urban outliers mask local differences among smaller settlements in mountainous regions. To account for this, settlements

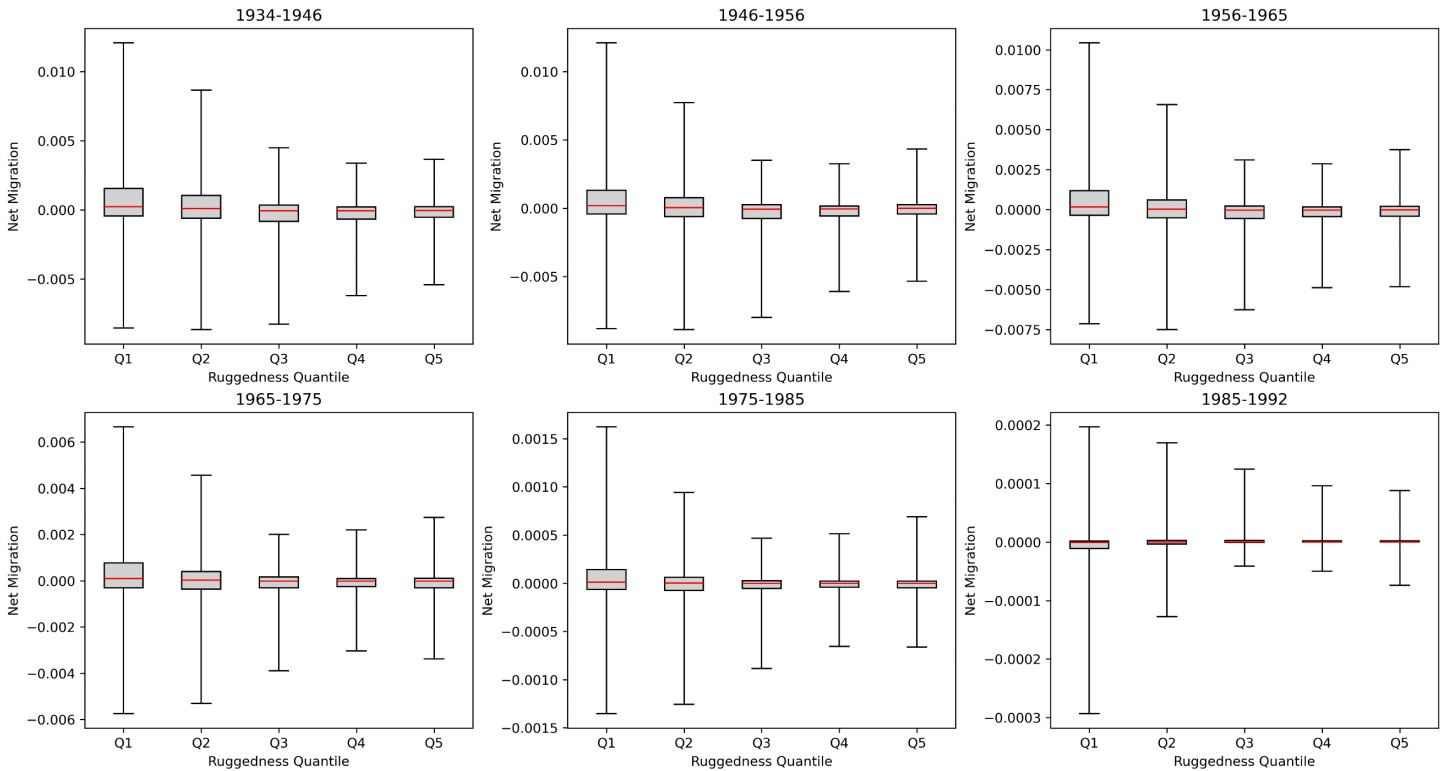

**Fig 8**. **Box plots per year pair, ranging from the 5th to the 95th percentile whiskers.** They highlight the TRI compared with net migration analysis at the settlement level. Settlements were divided into 5 equal quantile groups based on TRI.

were divided into five TRI quantile groups (Q1–Q5), as indicated in Fig 4. The box plots demonstrate that while median net migration values are negative across all groups, the highest ruggedness quantile (Q5) consistently records lower net migration compared to less rugged areas. A moderate group (Q3) shows slightly positive means, driven by strong inflows towards regional centres.

These findings highlight an important methodological consideration: absolute migration figures alone do not capture the relative demographic pressures experienced by smaller settlements in mountainous terrain. Since such settlements tend to have smaller population bases, even modest flows can represent substantial proportional change. Rather than normalising all flows by population, this study focuses on directional connections from highly rugged areas (Q5) to major urban destinations. Fig 9 illustrates this by mapping flows from the most rugged areas toward Plovdiv, showing how spatially constrained outmigration from mountain settlements feeds into regional urban growth.

Further temporal exploration reveals distinct urban destination preferences among migrants from rugged areas. Fig 10 traces the top destinations across intercensal intervals, showing consistent dominance by Sofia and Plovdiv, but with shifting importance of secondary cities such as Stara Zagora, Ruse, and Haskovo. This temporal variability suggests evolving regional hierarchies influenced by infrastructural development and industrialisation. Together, these results confirm that while rugged terrain restricts overall migration volumes, it also structures the geography of migration by channelling movement toward specific, accessible urban corridors.

Overall, the combined analysis of TRI, flow direction, and temporal destination patterns demonstrates that physical geography remains a central organising factor within Bulgaria's migration system. The approach provides a spatially explicit framework for examining how environmental constraints interact with demographic and infrastructural dynamics.

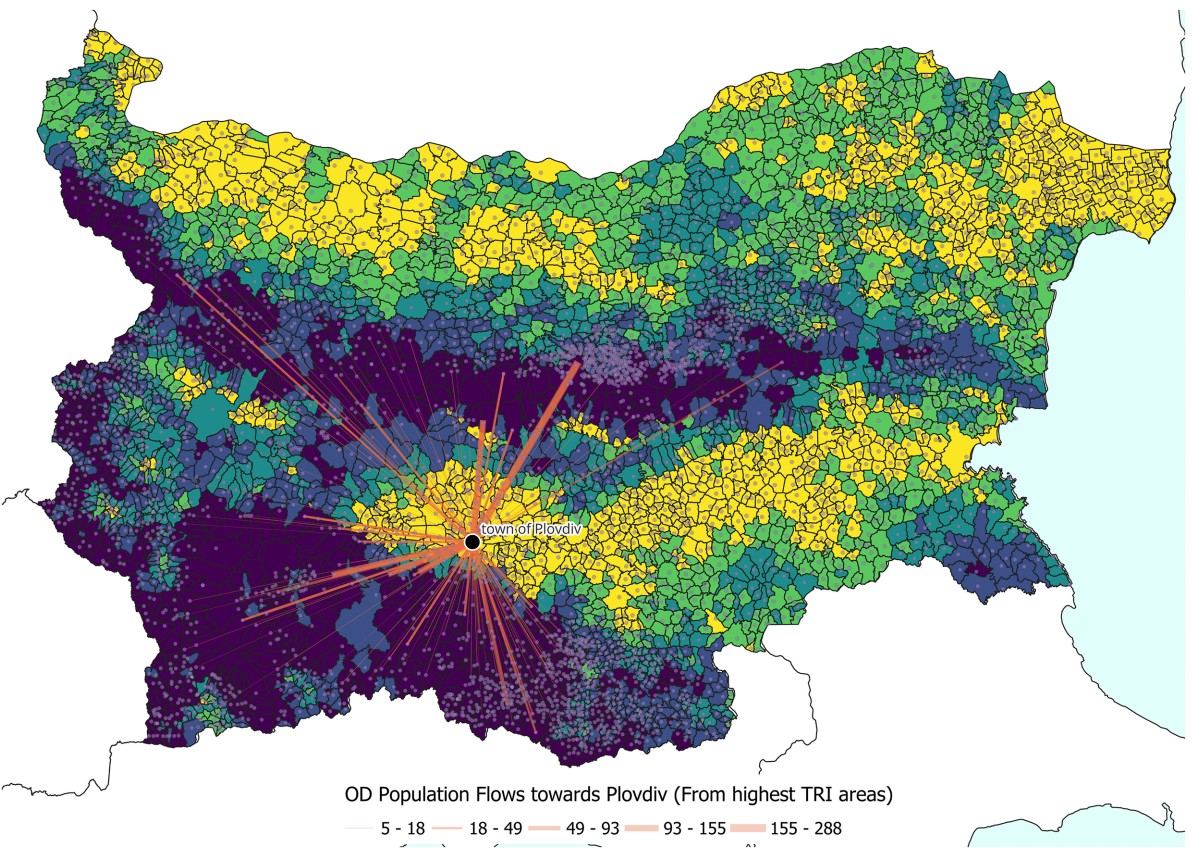

**Fig 9**. **Example showing the directional attraction of Plovdiv, compared with neighbouring settlements in the highest ruggedness quantile (Q5).**

While this study identifies clear associations, further research is required to quantitatively isolate the causal mechanisms underlying the observed temporal and regional variability.

## Discussion and limitations

The results of this study demonstrate that spatial interaction models can be used to effectively reproduce historical internal migration patterns in Bulgaria at both the settlement and regional levels. The settlement-level evaluation metrics show that the model performs robustly across intercensal intervals, even when applied to complex historical data, with MAPE ranging from 13% to 52%, MAE from 138 to 344, and SRMSE from 0.41 to 1.76. These values indicate consistent accuracy despite the large temporal span and varying quality of mid-twentieth-century census data. The higher MAPE values observed between 1965 and 1975 coincide with periods of major socio-economic restructuring, including rapid industrialisation and urban expansion, which the static model configuration cannot fully capture. Conversely, lower errors in 1934–1946 and 1985–1992 reflect a closer correspondence between predicted and observed demographic change. It should also be noted that SRMSE values exceeding 1 are likely influenced by a small number of high-magnitude outliers, particularly flows associated with the capital city, Sofia, which can disproportionately affect standardised error estimates.

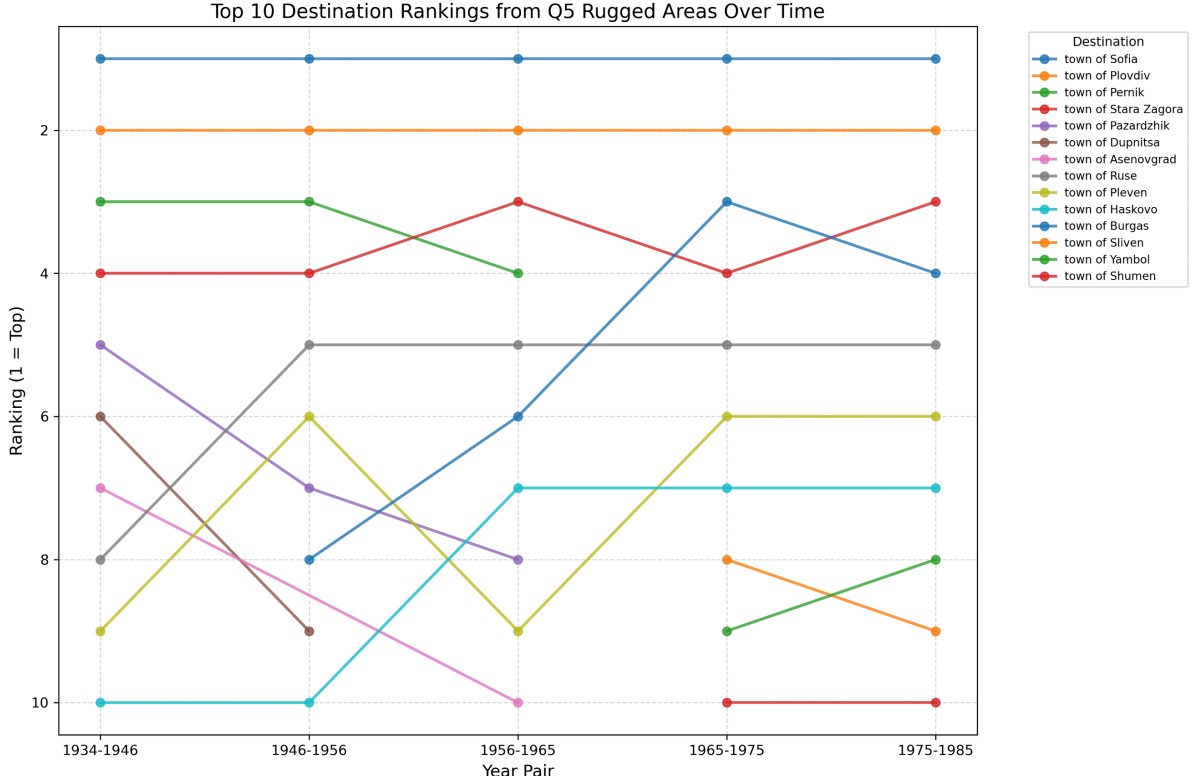

**Fig 10**. **Figure showing the top-ranking destinations of migrants of rugged or mountainous regions, per year pair.**

At the regional scale, modelled flows align closely with those reported by Philipov [5], showing an overall relative difference below 15% in total regional movements and correctly reproducing dominant interregional corridors. Discrepancies primarily occur in flows linked to Sofia and the Southwest, where rapid urban-industrial growth created atypically high migration volumes. These results confirm that the model captures both the magnitude and directionality of population movements, reinforcing the consistency of the settlement-level validation across spatial scales.

Beyond quantitative accuracy, the results offer new insights into how shifting socio-economic structures intersect with geographic accessibility in shaping migration decisions. The consistent relationship between improved infrastructure and inbound migration supports earlier observations on spatial inequality and accessibility in Bulgaria [1,3]. Our finding that a power-law decay function provides a better fit than an exponential one further aligns with established theories of spatial clustering and agglomeration [58], suggesting that individuals were willing to migrate over longer distances when economic or social opportunities justified the cost of movement.

The terrain analysis highlights the persistent role of physical geography in constraining population redistribution. Settlements in mountainous regions with high Terrain Ruggedness Index (TRI) values consistently show lower net migration and higher outflows relative to those in flatter areas, confirming earlier findings that geographic isolation limits access to services and employment [1,24]. Similar topographic effects have been reported in other mountainous regions [49], indicating that physical barriers to movement are a structural component of demographic change rather than a context-specific anomaly. While these results demonstrate clear spatial associations, further work is needed to quantify how such environmental constraints interact with local economic and infrastructural dynamics over time.

From a broader perspective, these findings contribute to ongoing debates about the spatial dimension of demographic change in Southeast Europe. They complement the regional demographic analyses by Daskalov et al. [59] and Dimou

et al. [60], showing that Bulgaria's historical migration system reflects both regionally specific conditions and broader spatial processes of urban concentration and peripheral decline. The settlement-level approach developed here thus provides a replicable framework for cross-national comparison, particularly in regions where fine-grained historical data are available.

## Limitations

Despite its strengths, our modelling framework has several limitations. First, the spatial interaction model (SIM) simplifies migration behaviour to a function of population and distance, excluding social, cultural, and political factors that likely influenced migration decisions during certain periods. Events such as industrial closures, collectivisation policies, or ethnic resettlements could have produced short-term deviations that are difficult to capture through a static functional form.

Second, although the use of a road network distance matrix improves accessibility estimates, it represents only a single dimension of travel cost. Historical conditions such as poor road maintenance, restricted car ownership, or seasonal closures, particularly in mountainous regions, are not fully represented in the OpenStreetMap-derived datasets. Extending the model to include railway connectivity [61], waterway routes, or evolving bus networks would yield a more comprehensive measure of historical accessibility.

Third, while this study focuses on internal migration, uncertainties remain in census-based population figures, particularly due to unrecorded migration and data inconsistencies between census years. These uncertainties may propagate through the flow estimates and affect model calibration. Future research could use multiple data sources, such as diaspora records or parish registers, to reduce bias in historical migration estimation, as discussed by Popek [62].

Finally, the interregional benchmark data from the 1976 Statistical Yearbook are possibly partially derived from the same census inputs used for model calibration, introducing some circularity in validation. Although this represents the best available source for the period, alternative approaches, such as cross-validation against independent archival or household-level datasets, would enhance the robustness of future analyses.

## Conclusion

This study demonstrates the capability of a Spatial Interaction Model (SIM) to reproduce Bulgaria's internal migration dynamics from 1934 to 1992 by integrating harmonised long-term census records with high-resolution road network data and external migration benchmarks. The inclusion of settlement-level evaluations using both relative and absolute error measures (MAPE, MAE, and SRMSE), together with flow-based validation against the 1976 interregional dataset, underlines the model's reliability across both local and macro-regional scales. The results corroborate previous research on Bulgaria's rural-to-urban transition and highlight the continuing influence of mountainous regions and physical geography in shaping migration flows.

By isolating migration processes from natural demographic change, the model provides refined estimates that clarify the interplay between population size, distance decay, and structural forces influencing relocation decisions. This approach offers a transferable framework for cross-regional analysis, particularly within Southeast Europe, where comparable historical and spatial datasets are increasingly available. Future studies could expand this framework by incorporating additional factors, such as historical rail connectivity or climatic variability, to further unpack the complexity of internal movement.

Ultimately, this work underscores the value of combining geospatial data science and historical demography to understand long-term population redistribution. A spatially explicit perspective not only deepens our understanding of Bulgaria's demographic history but also provides a robust empirical foundation for addressing present-day challenges of depopulation, regional imbalance, and sustainable development. With a clearer appreciation of how geography and infrastructure shape mobility, such insights can better support regional decision-making processes and contribute to the revitalisation of vulnerable local communities.

## Acknowledgments

The authors gratefully acknowledge the Bulgarian National Statistical Institute for providing census data, the Bulgarian Agency for Geodesy, Cartography, and Cadastre for essential geospatial information, and the developers of the Open Source Routing Machine (OSRM). Sincere thanks are also owed to the International Institute for Applied Systems Analysis (IIASA) for making the 1976 interregional flow data accessible and to colleagues who offered constructive commentary on earlier drafts of this work.

## Author contributions

**Conceptualization:** Petrus J. Gerrits, Guy Solomon, M. Erdem Kabadayi, Ana Basiri.

**Data curation:** Petrus J. Gerrits, Guy Solomon, M. Erdem Kabadayi, Ana Basiri.

**Formal analysis:** Petrus J. Gerrits, Guy Solomon, Ana Basiri.

**Funding acquisition:** M. Erdem Kabadayi, Ana Basiri.

**Investigation:** Petrus J. Gerrits, Guy Solomon.

**Methodology:** Petrus J. Gerrits, Guy Solomon, M. Erdem Kabadayi, Ana Basiri.

**Project administration:** Petrus J. Gerrits, Ana Basiri.

**Resources:** Petrus J. Gerrits, M. Erdem Kabadayi, Ana Basiri.

**Software:** Petrus J. Gerrits, Guy Solomon.

**Supervision:** M. Erdem Kabadayi, Ana Basiri.

**Validation:** Petrus J. Gerrits, Guy Solomon.

**Visualization:** Petrus J. Gerrits, Guy Solomon.

**Writing – original draft:** Petrus J. Gerrits.

**Writing – review & editing:** Petrus J. Gerrits, Guy Solomon, M. Erdem Kabadayi, Ana Basiri.

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
