## [Decision Letter · Decision Letter 0]

21 Aug 2025

PONE-D-25-23423

Mountain Moves: Spatial Interaction Modelling of Bulgaria’s Internal Migration (1934-1992)

PLOS ONE

Dear Dr. Gerrits,

Thank you for submitting your manuscript to PLOS ONE. After careful consideration, we feel that it has merit but does not fully meet PLOS ONE’s publication criteria as it currently stands. Therefore, we invite you to submit a revised version of the manuscript that addresses the points raised during the review process.

We look forward to receiving your revised manuscript.

Kind regards,

Yuxia Wang

Academic Editor

PLOS ONE

“The authors gratefully acknowledge the Bulgarian National Statistical Institute for providing census data, the Bulgarian Agency for Geodesy, Cartography, and Cadastre for essential geospatial information, and the developers of the Open Source Routing Machine (OSRM). Sincere thanks are also owed to the International Institute for Applied Systems Analysis (IIASA) for making the 1976 interregional flow data accessible and to colleagues who offered constructive commentary on earlier drafts of this work. Funding support was provided by the UrbanOccupationsOETR European Research Council funding (UrbanOccupationsOETR, grant agreement ID: 679097 and GeoAI LULC Seg, grant agreement: 101100837) and by the UK Research and Innovation Future Leaders Fellowships (grant reference: MR/Y011856/1 and MR/S01795X/2).”

“Funding support was provided by the UrbanOccupationsOETR European

Research Council (UrbanOccupationsOETR, grant agreement ID: 679097 and

GeoAI LULC Seg, grant agreement: 101100837, PI: M. Erdem Kabadayi), and by the UK Research and Innovation Future Leaders Fellowships (grant reference: MR/Y011856/1 and MR/S01795X/2, PI: Ana Basiri). The funders had no role in study design, data collection and analysis, decision to publish, or preparation of the manuscript.”

3. We note that Figures 1, 2, 3 and 8 in your submission contain [map/satellite] images which may be copyrighted. All PLOS content is published under the Creative Commons Attribution License (CC BY 4.0), which means that the manuscript, images, and Supporting Information files will be freely available online, and any third party is permitted to access, download, copy, distribute, and use these materials in any way, even commercially, with proper attribution. For these reasons, we cannot publish previously copyrighted maps or satellite images created using proprietary data, such as Google software (Google Maps, Street View, and Earth). For more information, see our copyright guidelines: http://journals.plos.org/plosone/s/licenses-and-copyright.

1. You may seek permission from the original copyright holder of Figures 1, 2, 3 and 8 to publish the content specifically under the CC BY 4.0 license.

4. Please remove your figures from within your manuscript file, leaving only the individual TIFF/EPS image files, uploaded separately. These will be automatically included in the reviewers’ PDF.

Reviewers' comments:

Reviewer's Responses to Questions

**Comments to the Author**

1. Is the manuscript technically sound, and do the data support the conclusions?

Reviewer #1: Yes

Reviewer #2: Yes

2. Has the statistical analysis been performed appropriately and rigorously?

Reviewer #1: Yes

Reviewer #2: No

3. Have the authors made all data underlying the findings in their manuscript fully available?

Reviewer #1: Yes

Reviewer #2: No

4. Is the manuscript presented in an intelligible fashion and written in standard English?

Reviewer #1: Yes

Reviewer #2: Yes

5. Review Comments to the Author

Reviewer #1: The paper focuses on population dynamics at a fine spatial resolution (settlement level) between 1934–1992, using geocoded census data and spatial interaction modeling. The study seeks to model spatial interactions which raises foundational questions like: Why do people move? What forces drive migration? A brief discussion or framing of key migration drivers at the beginning could strengthen the paper’s motivation. I find the overall approach and framing quite compelling. I have a few minor questions and suggestions for clarification below:

• Could the authors clarify the political context in Bulgaria during this period? Was the political environment relatively stable from 1934 to 1992? If not, how might periods of instability affect migration patterns?

• How frequently were censuses conducted in Bulgaria during this time frame? Including this information would help contextualize the data availability and temporal resolution of the analysis.

• Since population is a dynamic phenomenon, and the authors mention a major shift in 1990, I wonder if the final two years of the time frame are comparable to the earlier decades. Could this affect the stability or interpretability of the findings?

Line-by-Line Comments:

• Page 2, Line 35 – Great synthesis. The literature review is well-cited. However, could the authors more clearly state what specific gaps in prior studies this paper addresses?

• Page 5 – The term “friction distance” is used, but the rationale is unclear. Are migration flows assumed to decline with distance? If so, how is distance conceptualized, such as walking time, driving time, or geographic distance? Clarifying this would strengthen the interpretation.

• Page 7 – The authors mention using a Parquet database. It would be helpful to briefly explain why this format was chosen and how it benefits the analysis.

• Page 8, Lines 224–229 – This section is difficult to follow. Could the authors clarify the procedure or modeling step described here?

• Page 9 – The authors use standardized root mean square error (SRMSE) as a measure of model fit. There are some alternative goodness-of-fit metrics that could complement or strengthen the evaluation particular in spatial modeling context. IT would be good to support the use of SRMSE.

• Page 10, Line 285 – Figure 1?

• Page 10, Line 306 – Some of the unexpected results might stem from political or economic transitions, particularly around 1990. It would be helpful to acknowledge this possibility.

• Page 12, Line 322 – This is an important point: the analysis is based on spatial modeling, and spatial interactions matter. Please consider expanding on this, particularly how spatial context influences the observed dynamics.

Reviewer #2: Date: August 13, 2025

Dear Authors,

Thank you very much for conducting such an intriguing and timely study. While your manuscript “Mountain Moves: Spatial Interaction Modelling of Bulgaria’s Internal Migration (1934-1992)” is highly commendable, I recommend addressing the following key comments to enhance its overall quality and clarity. To facilitate further review, please clearly highlight all revisions in the updated manuscript.

Abstract:

• Could you please include the existing migration situation in Bulgaria so that the reader of your manuscript can have a full understanding of the situation and the necessity of your study for development activities, including policy advice or debts?

• No clear methods of data collection, processing, and analysis. Please make sure the abstract clearly outlines these methods.

• You need to minimize the number of keywords to 5-6.

Introduction

• Page 1, Lines 2-3: “Bulgaria’s demographics have undergone substantial shifts over the previous century.” Please support this statement with a review of literature and include the statistical values of change or shifts from/to and the annual rate of change.

• Page 3, Lines 82-92: Although you have clearly articulated the three key contributions of your study, the objectives remain unclear. Why?

• Page 4 Lines 93-100: Delete

Materials and methods

• Page 4, Lines 102-130: They are completely part of your data processing, and I don’t understand why you put them before the data acquisition. Please refine the structure of your data acquisition, processing, and analysis techniques. You are mixing things.

• Note: *** nothing is visible in all of your figures shown in the main body of the manuscript, not in the annex.

• This section needs major improvement, as the manuscript lacks clear descriptions of the scientific methods used for data acquisition, processing, analysis, and validation.

Results

• Page 12, Lines 338-367—Interdecadal variability of flows. I wonder if you could clearly depict the statistical variability and the triggering factors for the variability. The same is true for the regional and terrain migration dynamics section stated from lines 368-432.

• The results have not been validated. Please validate it.

• The quality of all your figures must be improved and saved into 300 dpi to enhance the visibility and ensure publication standards.

Discussion and Reflection

• Please include the findings when you discuss them. I didn’t see any statistical values discussed in your manuscript.

Conclusion

• There is no explicit advice or recommendation regarding policy debt.

• The conclusion does not explicitly state the limitations of your study. Could you please include them?

Thank you.

6. PLOS authors have the option to publish the peer review history of their article (what does this mean?). If published, this will include your full peer review and any attached files.

Reviewer #1: No

Reviewer #2: No

---

## [Author Response · Author response to Decision Letter 1]

7 Oct 2025

Please see the formatted pdf version enclosed

ID Comment Response

Journal requests:

J1 1. Please ensure that your manuscript meets PLOS ONE's style requirements, including those for file naming. The PLOS ONE style templates can be found at

Thank you. We have renamed the files in accordance with the guidelines

J2 2. We note that you have provided funding information that is currently declared in your Funding Statement. However, funding information should not appear in the Acknowledgments section or other areas of your manuscript. We will only publish funding information present in the Funding Statement section of the online submission form.

“Funding support was provided by the UrbanOccupationsOETR European

Research Council (UrbanOccupationsOETR, grant agreement ID: 679097 and

GeoAI LULC Seg, grant agreement: 101100837, PI: M. Erdem Kabadayi), and by the UK Research and Innovation Future Leaders Fellowships (grant reference: MR/Y011856/1 and MR/S01795X/2, PI: Ana Basiri). The funders had no role in study design, data collection and analysis, decision to publish, or preparation of the manuscript.”

Thank you, funding statements were removed from the Acknowledgement section and only present in the funding statement section. The Funding Statement has remained the same as provided earlier.

J3 3. We note that Figures 1, 2, 3 and 8 in your submission contain [map/satellite] images which may be copyrighted. All PLOS content is published under the Creative Commons Attribution License (CC BY 4.0), which means that the manuscript, images, and Supporting Information files will be freely available online, and any third party is permitted to access, download, copy, distribute, and use these materials in any way, even commercially, with proper attribution. For these reasons, we cannot publish previously copyrighted maps or satellite images created using proprietary data, such as Google software (Google Maps, Street View, and Earth). For more information, see our copyright guidelines: http://journals.plos.org/plosone/s/licenses-and-copyright

See further details in the comments

Thank you. All of the sources for the satellite imagery were checked (we only use OSM, free access sources) and adjusted according to the guidelines.

J4 4. Please remove your figures from within your manuscript file, leaving only the individual TIFF/EPS image files, uploaded separately. These will be automatically included in the reviewers’ PDF.

These have been removed.

J5 If the reviewer comments include a recommendation to cite specific previously published works, please review and evaluate these publications to determine whether they are relevant and should be cited. There is no requirement to cite these works unless the editor has indicated otherwise.

Checked

ID Comment Response

Reviewer 1:

R1.1 Could the authors clarify the political context in Bulgaria during this period? Was the political environment relatively stable from 1934 to 1992? If not, how might periods of instability affect migration patterns?

On page 3, line 54, I revised "By focusing on the period from 1934 to 1992, our study captures an era of relative administrative, census-based consistency" to "Our study focuses on the period from 1934 to 1992, an era characterized by administrative consistency in census taking and relative political stability, including the immediate periods before and after the Communist rule in Bulgaria."

R1.2 How frequently were censuses conducted in Bulgaria during this time frame? Including this information would help contextualize the data availability and temporal resolution of the analysis.

On page 2, line 37-38, I revised "Our period of examination is based on a purposeful selection of population censuses and extends between 1934 and 1992." to "The period of our examination is 1934 to 1992, and it is based on all seven population censuses conducted within these years."

R1.3 Since population is a dynamic phenomenon, and the authors mention a major shift in 1990, I wonder if the final two years of the time frame are comparable to the earlier decades. Could this affect the stability or interpretability of the findings?

Yes, we agree that the last intercensal period in our research (following 1985 - 1992) is a exceptional period in Bulgarian history. However, this does not have an impact the stability of our spatial interaction analysis, as it investigates each intercensal period independently. We have updated the manuscript to make this clearer and added it in the discussion section.

R1.4 Page 2, Line 35 – Great synthesis. The literature review is well-cited. However, could the authors more clearly state what specific gaps in prior studies this paper addresses?

Thank you for the feedback. We have changed the wording to highlight the gaps that prior studies in this paper have missed. Our contribution is the spatial-temporal construction of a spatial interaction model that highlights the movement of people during these census periods. We have updated the language and structure of the to make this clearer. To date, nobody has provided a scalable solution to a national issue. This data can provide an additional usable source of reference for further investigation.

R1.5 Page 5 – – The term “friction distance” is used, but the rationale is unclear. Are migration flows assumed to decline with distance? If so, how is distance conceptualized, such as walking time, driving time, or geographic distance? Clarifying this would strengthen the interpretation.

Thanks for this question. There are two elements here:

1. Typically, SIMs assume that the strength of relationships between places will decline over distance (all else equal). However, this must be calibrated for each model, to best fit the characteristics of the spatial system. How we have fitted the most appropriate distance decay function is discussed in the analysis section.

2. We measure distance according to the metres between two places (as recorded by the OSM road network), as there is no historical transport network data for this period. We consider this to be a better proxy for distance travelled in migration than Euclidean distance (which we compare in Figure 3). This is particularly important, given the latter does not account for terrain (as discussed in the Data Processing section. We have updated the language and structure of the to make this clearer.

R1.6 Page 7 – The authors mention using a Parquet database. It would be helpful to briefly explain why this format was chosen and how it benefits the analysis.

The parquet database been used as a pipeline which enables the parameter optimisation process. This database has made it possible to investigate the million edges (lines between the origin and destinations) in our network. Parquet databases are a recent geospatial development and one that makes large spatial analysis quicker, more accessible and sharing data easier.

The usage of this database has been better positioned also in the flow chart on figure 2. At the beginning of the material and methods section.

R1.7 Page 8, Lines 224–229 – This section is difficult to follow. Could the authors clarify the procedure or modeling step described here?

The methods and materials section has been extensively reconstructed based on the comments of both reviewers and has now been given a more clear structure (data acquisition, data processing, analysis, validation, followed by results).

R1.8 Page 9 – The authors use standardized root mean square error (SRMSE) as a measure of model fit. There are some alternative goodness-of-fit metrics that could complement or strengthen the evaluation particular in spatial modeling context. It would be good to support the use of SRMSE.

We agree that goodness-of-fit can be evaluated with several complementary measures, and for this reason, in addition to SRMSE we also report MAE and MAPE in the manuscript. We emphasised SRMSE because it provides a scale-independent assessment of predictive accuracy that facilitates comparison across different census periods and settlement sizes—an especially useful property in spatial interaction modelling where population magnitudes vary widely. To strengthen the manuscript, we have clarified this rationale and highlighted how SRMSE complements absolute (MAE) and relative (MAPE) error measures in our evaluation framework. Additionally, we have added references to other SIM papers that have used this measurement for model fit, as well as its availability in the SpInt Python Library Package.

R1.9 Page 10, Line 285 – Figure 1?

We refer to figure 1 for the reader unfamiliar with the area to see the distinct 7 aggregated regions. These regions are indicated using the yellow-ringed labels

R1.10

Page 10, Line 306 – Some of the unexpected results might stem from political or economic transitions, particularly around 1990. It would be helpful to acknowledge this possibility. Thank you for this suggestion and yes, we agree that for this intercensal period, the political transition is a likely cause for this difference with previous periods. We have added this as a possibility in the text.

R1.11

Page 12, Line 322 – This is an important point: the analysis is based on spatial modeling, and spatial interactions matter. Please consider expanding on this, particularly how spatial context influences the observed dynamics. Thank you for your question, we have expanded on this point and have included more context on how this is related to our observed migration dynamics.

ID Comment Response

Reviewer 2

R2.1 Abstract: Could you please include the existing migration situation in Bulgaria so that the reader of your manuscript can have a full understanding of the situation and the necessity of your study for development activities, including policy advice or debts?

Thank you. We have added more context to the current situation in Bulgaria, and have added the following to the abstract:

"In a country that has experienced dramatic rural decline alongside rapid urban growth, our study offers essential historical context to inform contemporary policy debates and urban planning initiatives." to "Although our study does not offer direct policy advice, it provides essential historical context for contemporary policy debates and urban planning initiatives in a country that has experienced both significant rural decline and rapid urbanization." in the abstract.

R2.2 Abstract: No clear methods of data collection, processing, and analysis. Please make sure the abstract clearly outlines these methods.

This has been added to the abstract, as well as better structured in the methods and material section.

R2.3 Abstract: Minimize keywords to 5–6.

Keywords have been minimized to the following: spatial interaction model, internal migration, historical census, rural depopulation, settlement-level analysis, topography

R2.4 Introduction (Page 1, Lines 2-3): “Bulgaria’s demographics have undergone substantial shifts over the previous century.” Please support this statement with a review of literature and include the statistical values of change or shifts from/to and the annual rate of change.

Thank you for this suggestion. We have added more literature that discusses the demographic development in the previous century, such as:

38 - Chavdar Mladenov, Emil Dimitrov, and Boris Kazakov. “Demographical development of Bulgaria during the transitional period”. In:

58 and 59 - Nickolay Tsekov. “Rural depopulation and the changing scope of the rural settlement network in Bulgaria in 1946-201”. In: Papers of BAS 4

Statistical values of changes are not added in the opening paragraph, but to provide background and scope, we have added the overall demographic change between lines 72 and 87.

R2.5 Introduction (Page 3, Lines 82–92): Although you have clearly articulated the three key contributions of your study, the objectives remain unclear. Why?

We revised the paragraph on study contributions to clarify the main research objective and its relationship to the paper’s methodological and empirical contributions. The updated paragraph now states that the study provides a spatially explicit account of internal migration in Bulgaria at the settlement level, capturing both the temporal evolution of migration intensity and the influence of topographic and regional constraints. Furthermore, to support transparency and reproducibility, all data, code, and figures used in this analysis are publicly available and referenced in the Data Availability statement.

R2.6 Introduction (Page 4, Lines 93–100): Delete.

deleted

R2.7 Methods (Page 4, Lines 102-130): They are completely part of your data processing, and I don’t understand why you put them before the data acquisition. Please refine the structure of your data acquisition, processing, and analysis techniques. You are mixing things.

Thank you for this feedack and we acknowledge this section needed major work. The structure of the materials and methods section has been completely reconsidered, and a flowchart has been added to the beginning of the section to illustrate the research steps and data collection process.

R2.8 Figures: nothing is visible in figures. Improve figure quality.

Figures have been improved to all be at the minimum of 300 dpi. We have excluded the figures from the manuscript, following the journal guidelines. Therefore we hope they are suitably visible. In case this is due to the upload and sharing process. We believe the PDF version also has a download link in the top right corner for a better quality download.

R2.9 Methods: This section needs major improvement, as the manuscript lacks clear descriptions of the scientific methods used for data acquisition, processing, analysis, and validation.

Thank you for this feedback. As mentioned in earlier comment, we have revised this section significantly;

1) By restructuring the section in clear sections (data acquisition, processing, analysis, and validation).

2) By including a flow chart of the entire workflow in the beginning of this section. The workflow describes the steps taken from the data acquisition to the validation and results section.

R2.10 Results: (Page 12, Lines 338–367): Interdecadal variability of flows. I wonder if you could clearly depict the statistical variability and the triggering factors for the variability. The same is true for the regional and terrain migration dynamics section stated from lines 368-432.

We revised the results section on intercensal variability to clarify the statistical interpretation of migration flow distributions and to explicitly address interdecadal variability. The updated text now emphasises that the observed variability reflects the combined influence of demographic, economic, and infrastructural dynamics, while acknowledging that these individual drivers cannot be isolated within the current modelling framework.

A new comment has also been added explaining that the dataset provides a platform for future research aimed at disentangling such triggers and examining potential relationships between intercensal periods, whether for causal inference, counterfactual analysis, or for understanding broader migration trends. This clarification reinforces the connection between model outcomes, statistical variability, and Bulgaria’s broader historical demographic context.

R2.11 Results (Page 12, Lines 368–432): Clarify regional and terrain migration dynamics.

We clarified the Terrain Ruggedness (TRI) Effects section to better explain how topography influences migration patterns. The revision now links terrain ruggedness to regional variability in migration intensity and destination choice.

R2.12 Result

---

## [Decision Letter · Decision Letter 1]

4 Jan 2026

Mountain Moves: Spatial Interaction Modelling of Bulgaria’s Internal Migration (1934-1992)

PONE-D-25-23423R1

Dear Dr. Gerrits,

We’re pleased to inform you that your manuscript has been judged scientifically suitable for publication and will be formally accepted for publication once it meets all outstanding technical requirements.

Kind regards,

Ye Wei, PhD

Academic Editor

PLOS One

Additional Editor Comments (optional):

Reviewers' comments:

Reviewer's Responses to Questions

**Comments to the Author**

1. If the authors have adequately addressed your comments raised in a previous round of review and you feel that this manuscript is now acceptable for publication, you may indicate that here to bypass the “Comments to the Author” section, enter your conflict of interest statement in the “Confidential to Editor” section, and submit your "Accept" recommendation.

Reviewer #1: All comments have been addressed

Reviewer #3: All comments have been addressed

2. Is the manuscript technically sound, and do the data support the conclusions?

Reviewer #1: Yes

Reviewer #3: Yes

3. Has the statistical analysis been performed appropriately and rigorously?

Reviewer #1: Yes

Reviewer #3: Yes

4. Have the authors made all data underlying the findings in their manuscript fully available?

Reviewer #1: Yes

Reviewer #3: Yes

5. Is the manuscript presented in an intelligible fashion and written in standard English?

Reviewer #1: Yes

Reviewer #3: Yes

6. Review Comments to the Author

Reviewer #1: Thanks for addressing all the comments that I have pointed out earlier. Just one tiny correction, "limitation" title needs to be title case, currently it is all lower case.

Reviewer #3: (No Response)

7. PLOS authors have the option to publish the peer review history of their article (what does this mean?). If published, this will include your full peer review and any attached files.

Reviewer #1: No

Reviewer #3: No

---

## [Editor Report · Acceptance letter]

PONE-D-25-23423R1

PLOS One

Dear Dr. Gerrits,

I'm pleased to inform you that your manuscript has been deemed suitable for publication in PLOS One. Congratulations! Your manuscript is now being handed over to our production team.

Kind regards,

on behalf of

Dr. Ye Wei

Academic Editor

PLOS One